# $k$-Means Clustering of Lines for Big Data

**Yair Marom**
Department of Computer Science
University of Haifa
Haifa, Israel
`yairmrm@gmail.com`

**Dan Feldman**
Department of Computer Science
University of Haifa
Haifa, Israel
`dannyf.post@gmail.com`

## Abstract

The input to the *k-mean for lines* problem is a set $L$ of $n$ lines in $\mathbb{R}^d$, and the goal is to compute a set of $k$ centers (points) in $\mathbb{R}^d$ that minimizes the sum of squared distances over every line in $L$ and its nearest center. This is a straightforward generalization of the $k$-mean problem where the input is a set of $n$ points instead of lines.

We suggest the first PTAS that computes a $(1 + \varepsilon)$-approximation to this problem in time $O(n \log n)$ for any constant approximation error $\varepsilon \in (0, 1)$, and constant integers $k, d \geq 1$. This is by proving that there is always a weighted subset (called coreset) of $dk^{O(k)} \log(n)/\varepsilon^2$ lines in $L$ that approximates the sum of squared distances from $L$ to *any* given set of $k$ points.

Using traditional merge-and-reduce technique, this coreset implies results for a streaming set (possibly infinite) of lines to $M$ machines in one pass (e.g. cloud) using memory, update time and communication that is near-logarithmic in $n$, as well as deletion of any line but using linear space. These results generalized for other distance functions such as $k$-median (sum of distances) or ignoring farthest $m$ lines from the given centers to handle outliers.

Experimental results on 10 machines on Amazon EC2 cloud show that the algorithm performs well in practice. Open source code for all the algorithms and experiments is also provided.

## 1 Introduction

### 1.1 Background

Clustering is the task of partitioning the input set to subsets, where items in the same subset (cluster) are similar to each other, compared to items in other clusters. There are many different clustering techniques, but arguably the most common in both industry and academy is the $k$-mean problem, where the input is a set $P$ of $n$ points in $\mathbb{R}^d$, and the goal is to compute a set $C$ of $k$ centers (points) in $\mathbb{R}^d$, that minimizes the sum of squared distances over each point $p \in P$ to its nearest center in $C$, i.e.

$$C \in \underset{C' \subseteq \mathbb{R}^d, |C'|=k}{\arg \min} \sum_{p \in P} \min_{c' \in C'} \|p - c'\|^2.$$

A very common heuristics to solve this problem is the Lloyd's algorithm [3, 22], that is similar to the EM-Algorithm that is described in Section 5 in supplementary material [2]. We consider a natural generalization of this $k$-mean problem, where the input set $P$ of $n$ points is replaced by a set $L$ of $n$ lines in $\mathbb{R}^d$; See Fig. 2. Here, the distance from a line to a center $c$ is the closest Euclidean distance to $c$ over all the points on the line. Since we only assume the "weak" triangle inequality between

points, our solution can easily be generalized to sum of distances to the power of any constant $z \geq 1$ as explained e.g. in [7, 6].

**Motivation** for solving the $k$-line mean problem arises in many different fields, when there is some missing entry in all or some of the input vectors, or incomplete information such as a missing sensor. For example, a common problem in Computer Vision is to compute the position of a point or $k$ points in the world, based on their projections on a set of $n$ 2D-images, which turn into $n$ lines via the pinhole camera model; See Fig 1, and [18, 27] for surveys.

In Data Science and matrix approximation theory, every missing entry turns a point (database's record) into a line by considering all the possible values for the missing entry. E.g., $n$ points on the plane from $k$-mean clusters would turn into $n$ horizontal/vertical lines that intersect "around" the $k$-mean centers. The resulting problem is then $k$-mean for $n$ lines [24, 25]. One can consider also an applications to semi-supervised learning - $k$-mean for mixed points and lines. This problem arises when lines are unlabeled points (last axis is a label) and we want to add a label to the farthest lines from the points.

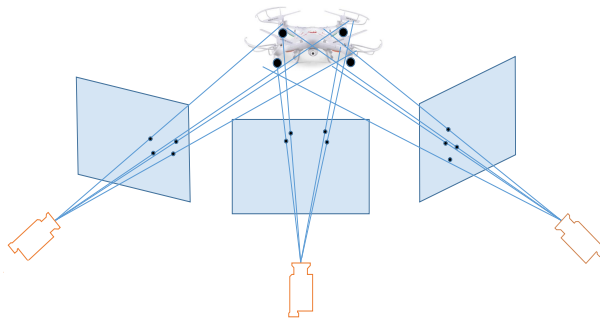

Figure 1: Application of k-line mean for computer vision. Given a drone (or any other rigid body) that is captured by $n$ cameras - our goal is to locate the 3 dimensional position of the drone in space by identifying $k = 4$ fixed known markers/features on the drone. Each point on each image corresponds to a line that passes through it and the pin-hole of the camera. Without noise, all the lines intersect at the same point (marker in $\mathbb{R}^3$). Otherwise their 1-mean is a natural approximation.

## 1.2 Related Work

The $k$-mean problem and its variance has been researched in numerous papers over the recent decades, especially in the machine learning community , see [16, 20, 28, 5] and references therein. There are also many results regarding projective clustering, when the $k$ centers are replaced by lines or $j$-dimensional subspaces instead of points.

However, significantly less results are known for the case of clustering subspaces, or even lines. A possible reason might be to the fact that the triangle inequality or its weaker version holds for a pair of points but not for lines, even in the planar case: two parallel lines can have an arbitrarily large distance from each other, but still intersect with a third line simultaneously. Gao, Langebreg and Schulman [12] used Helly's theorem [8] (intersection of convex sets) to introduce the "$k$-center problem" for lines, that aims to cover a collection of lines by the smallest $k$ balls in $\mathbb{R}^3$.

Langebreg and Schulman [11] addressed the 1-convex sets center problem that aims to compute a ball that intersects a given set of $n$ convex sets such as lines and $\Delta$-affine subspaces. This type of non-clustering problems ($k = 1$) is easier since it admits a convex optimization problem instead of a clustering non-convex problem.

Unlike the case of numerous theoretical papers that study the $k$-mean problem for points, we did not find any provable solution for the case of $k$-line mean problem or even an efficient PTAS (polynomial approximation scheme). There are very few related results that we give here. A solution for the special case of $d = 2$ and sum of distances was considered in [23]. Lee and Schulman [17] studied the generalization of the $k$-center problem (maximum over the $n$ distances, instead of their sum), for the case where the input is a set of $n$ affine subspaces, each of dimension $\Delta$. In this case, the size of

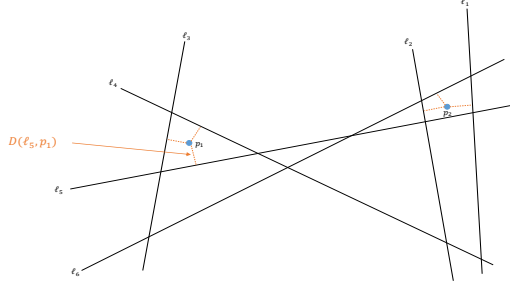

Figure 2: Problem Statement demonstration for the planar case. The input is a set $L = \{\ell_1, \ldots, \ell_6\}$ of $n = 6$ lines in $\mathbb{R}^2$, and our goal is to find the $k = 2$ points (centers) $p_1, p_2 \in \mathbb{R}^2$ that minimize the sum of Euclidean distances from each center to its nearest line in $L$. See Section 2.1 definition of the distance function $D$.

the coreset is exponential in $d$, which was proved to be unavoidable even for a coreset of 1-center (single point) for this type of covering problems. The corresponding covering problems can then be solved using traditional computational geometry techniques.

Table 1 summarizes the above results.

| Problem | Running Time | Approx. Factor | Paper |
|---|---|---|---|
| 1-line center in $\mathbb{R}^d$ | $nd(1/\varepsilon)^{O(1)}$ | $1 + \varepsilon$ | [11] |
| 2-line centers in $\mathbb{R}^2$ | $O\left(n \log\left(1/\varepsilon\right)\left(d + \log n\right)\right)$ | $2 + \varepsilon$ | [12] |
| 3-line centers in $\mathbb{R}^2$ | $O\left(nd \log(1/\varepsilon) + \frac{n \log^2(n) \log(1/\varepsilon)}{\varepsilon}\right)$ | $2 + \varepsilon$ | [12] |
| 1-center for convex-sets in $\mathbb{R}^d$ | $O\left(n^{\Delta+1} d(1/\varepsilon)^{O(1)}\right)$ | $1 + \varepsilon$ | [11] |
| $k$-line median in $\mathbb{R}^3$ | iterative, unbounded | unbounded | [21] |
| $k$-line median in $\mathbb{R}^2$ | $n\left(\frac{\log n}{\varepsilon}\right)^{O(k)}$ | $1 + \varepsilon$ | [23] |
| 1-$\Delta$-flats center in $\mathbb{R}^d$ | $O\left(\frac{nd\Delta}{\varepsilon^2} \log \frac{\Delta}{\varepsilon}\right)$ | $1 + \varepsilon$ | [17] |
| 2-$\Delta$-flats centers in $\mathbb{R}^d$ | $O\left(dn^2 \log n\right)$ | $O\left(\Delta^{1/4}\right)$ | [17] |
| 3-$\Delta$-flats centers in $\mathbb{R}^d$ | $2^{O(\Delta(1+1/\varepsilon^2))} nd$ | $1 + \varepsilon$ | [17] |
| $k$-$\Delta$-flats centers in $\mathbb{R}^d$ | $2^{O(\Delta k \log k(1+1/\varepsilon^2))} nd$ | $1 + \varepsilon$ | [17] |
| $k$-line median in $\mathbb{R}^d$ | $O\left(d^3 n \log(n) k \log k + (d/\varepsilon)^2\right) + ndk^{O(k)}$ | $1 + \varepsilon$ | **Our** |

Table 1: Summary of related results for $k$ centers of $n$ points in $\mathbb{R}^d$. The dimension of the quarried subspaces is denoted by $\Delta$ and the error rate is by $\varepsilon$.

### 1.3 Main Contribution

Our main technical result is an algorithm that gets a set $L$ of $n$ lines in $\mathbb{R}^d$, an integer $k \geq 1$, and computes an $\varepsilon$-coreset (see Definition 2.7) of size $dk^{O(k)} \log(n)/\varepsilon^2$ for $L$ and every error parameter $\varepsilon > 0$, in a near-linear running time in the number of data lines $n$, and polynomial in the dimensionality $d$ and the number $k$ of desired centers; See Theorem 2.8 for details and exact bounds. Using this coreset with a merge-and-reduce technique, we achieve the following results:

1. An algorithm that, during one pass, maintains and outputs a $(1 + \varepsilon)$-approximation to the $k$-line mean of the lines seen so far; See Definition 2.4.

2. A streaming algorithm that computes an $(1+\varepsilon)$-approximation for the $k$-line mean of a set $L$ of lines that may be distributed (partitioned) among $M$ machines, where each machine needs to send only $d^3 k^{O(k)} \log^2 n$ input lines to the main server at the end of its computation.

3. Experimental results on 10 machines on Amazon EC2 Cloud [9] show that the algorithm performs well in practice, boost the performance of existing EM-heuristic [19].

**Why the coreset is exponential in $k$?** The worst case size of our coreset is polynomial in $\log n$ but exponential in $k$. Still, it is tight: such lower bounds are known for weighted centers [14] which is a special case of coreset for lines, as explained in [10]. Fortunately, it seems that it is only a theoretical worst-case bound for a very tailored and artificial synthetic example. Our experiments imply that the error on real-world data sets is far better than our theorem predicts *using the same importance distribution* but for a smaller sample size. In addition, we can partition the $n$-line set in the hope that each subset would be served by at most $k$ points.

## 2 Problem Statement and Theoretical Result

### 2.1 Preliminaries and Problem Statement

For an integer $n \geq 1$ we define $[n] = \{1, \ldots, n\}$. From here and in the rest of the paper, we assume that we are given a function $D : \mathbb{R}^d \times \mathbb{R}^d \to \mathbb{R}$ and a constant $\rho > 0$ such that $D(a, b) \leq \rho(D(a, c) + D(c, b))$ for every $a, b, c \in \mathbb{R}^d$.

**Definition 2.1 (weighted set)** *A weighted set of lines is a pair $L' = (L, w)$ where $L$ is a set of lines in $\mathbb{R}^d$, and $w : L \to (0, \infty)$ is a function that maps every $\ell \in L$ to $w(\ell) \geq 0$, called the weight of $\ell$. A weighted set $(L, 1)$ where $1$ is the weight function $w : L \to \{1\}$ that assigns $w(\ell) = 1$ for every $\ell \in L$ may be denoted by $L$ for short.*

**Definition 2.2 (distance)** *The Euclidean distance between a pair of points is denoted by the function $D : \mathbb{R}^d \times \mathbb{R}^d \to [0, \infty)$, s.t. for every $x, y \in \mathbb{R}^d$ we have $D(x, y) = \|x - y\|_2$. For every set $X \subseteq \mathbb{R}^d$ and a point $x \in \mathbb{R}^d$, we define the distance from $X$ to $x$ by $D(X, x) = \inf_{q \in X} D(q, x)$, and for every set $Y \subseteq \mathbb{R}^d$, we denote the distance from $X$ to $Y$ by $D(X, Y) = \inf_{(x,y) \in X \times Y} D(x, y)$.*

**Definition 2.3 (cost)** *For every set $P \subseteq \mathbb{R}^d$ of $k$ points and a weighted set of lines $L' = (L, w)$ in $\mathbb{R}^d$, we denote the sum of weighted distances from $L$ to $P$ by $\mathrm{cost}(L', P) = \sum_{\ell \in L} w(\ell) D(\ell, P)$.*

A natural generalization of the $k$-mean problem is to replace the input set of points $P$ by a set $L$ of $n$ lines in $\mathbb{R}^d$.

**Definition 2.4 ($k$-mean for lines)** *Let $L' = (L', w)$ be a weighted set of lines in $\mathbb{R}^d$ and $k \geq 1$ be an integer. A set $P^* \subseteq \mathbb{R}^d$ is a $k$-mean of $L'$ if it minimizes $\mathrm{cost}(L', P)$ over every set $P$ of $k$ points in $\mathbb{R}^d$.*

### 2.2 Theoretical Results

We begin with our first result - the first bicriteria approximation for the general $k$-mean of lines in $\mathbb{R}^d$, for any integers $k, d \geq 1$.

**Definition 2.5 ($\alpha, \beta$-approximation)** *Let $L$ be a finite set of lines in $\mathbb{R}^d$, $k \geq 1$ be an integer and $P^* \subseteq \mathbb{R}^d$ be a $k$-mean of $L$; See Definition 2.4. Then, for every $\alpha, \beta \geq 0$, a set $B \subseteq \mathbb{R}^d$ of $k\beta$ points is called $(\alpha, \beta)$-approximation of $L$, if*

$$\mathrm{cost}(L, B) \leq \alpha \cdot \mathrm{cost}(L, P^*).$$

*If $\beta = 1$ then $B$ is called an $\alpha$-approximation of $L$. , if $\alpha = \beta = 1$ then $B$ is called the optimal solution.*

**Theorem 2.6** *Let $L$ be a set of $n$ lines in $\mathbb{R}^d$, $k \geq 1$ be an integer, $\delta \in (0, 1)$ and*

$$m \geq c \left( dk \log_2 k + \log_2 \left( \frac{1}{\delta} \right) \right),$$

*for a sufficiently large constant $c > 1$ that can be determined from the proof. Let $B$ be the output set of a call to* BI-CRITERIA-APPROXIMATION$(L, m)$; *See Algorithm 2. Then,*

$$|B| \in O\left(\log n \left(dk \log k + \log(1/\delta)\right)\right) \qquad (1)$$

*and with probability at least $1 - \delta$,*

$$\mathrm{cost}(L, B) \leq 4\rho^2 \cdot \min_{P \subseteq \mathbb{R}^d, |P|=k} \mathrm{cost}(L, P).$$

*Moreover, $B$ can by computed in $O\left(nd^2 k \log k + m^2 \log n\right)$ time.*

Coreset is a problem dependent data summarization. The definition of coreset is not consistent among papers. In this paper, the input is usually a set of lines in $\mathbb{R}^d$, but for the streaming case we compute coreset for union of (weighted) coresets and thus weights will be needed. We use the following definition of Feldman and Kfir [15].

**Definition 2.7 ($\varepsilon$-coreset [15])** *For an approximation error $\varepsilon > 0$, the weighted set $S' = (S, u)$ is called an $\varepsilon$-coreset for the query space $(P', Y, f, \mathrm{loss})$, if $S \subseteq P, u : S \to [0, \infty)$, and for every $y \in Y$ we have*

$$(1 - \varepsilon) f_{\mathrm{loss}}(P', y) \leq f_{\mathrm{loss}}(S', y) \leq (1 + \varepsilon) f_{\mathrm{loss}}(P', y).$$

**Theorem 2.8 (coreset for $k$-line mean)** *Let $L$ be a set of $n$ lines in $\mathbb{R}^d$, $k \geq 1$ be an integer, $\varepsilon, \delta \in (0, 1)$ and $m > 1$ be an integer such that*

$$m \geq \frac{cd^2 k \log^2(k) \log^2(n) + \log(1/\delta)}{\varepsilon^2},$$

*for some universal constant $c > 0$, and $\mathcal{Q}_k = \left\{B \subseteq \mathbb{R}^d \mid |B| = k\right\}$. Let $(S, u)$ be the output of a call to* CORESET$(L, k, m)$; *see Algorithm 4. Then, with probability at least $1 - \delta$, $(S, u)$ is an $\varepsilon$-coreset for the query space $(L, \mathcal{Q}_k, D, \|\cdot\|_1)$. Moreover, $(S, u)$ can be computed in time*

$$O\left(d^3 n \log(n) k \log k + (d/\varepsilon)^2\right) + ndk^{O(k)}.$$

Using Kfir and Feldman coreset on streaming framework [15], we enable to boost performance and instead of performing the coreset calculation on the whole data in one piece - we performed the action on batches we read one after the other (the streaming version), and its validity can be seen in the carried out experiments in Section 4.

## 3 Algorithms and Technique

We present here the first bi-criteria solution for the $k$-line mean problem, where Alg. 1 is a sub-procedure that is being called during the running of Alg. 2.

---

**Algorithm 1:** CENTROID-SET$(L)$

**Input:**     A finite set $L$ of $n$ lines in $\mathbb{R}^d$.
**Output:**   A set $G \subseteq \mathbb{R}^d$ of $O(n^2)$ points.

**1** **for** *every $\ell \in L$* **do**
**2**    **for** *every $\ell' \in L \setminus \ell$* **do**
**3**       Compute $q(\ell, \ell') \in \arg\min_{x \in \ell} D(\ell', x)$
        // the closest point on $\ell$ to $\ell'$. Ties broken arbitrarily.
**4**    $Q(\ell) := \{q(\ell, \ell') \mid \ell' \in L \setminus \{\ell\}\}$
**5**    $G := \bigcup_{\ell \in L} Q(\ell)$
**6** **return** $G$

---

**Overview of Algorithm 2.** The input to the algorithm is a set $L$ consist of $n$ lines in $\mathbb{R}^d$ and a positive integer $m \geq 1$. In each iteration of the algorithm it picks a small uniform sample $S$ of the input in Line 3, compute their centroid-set $G$ using a call to Algorithm 1 in Line 4, and add them to the output set $B$ in Line 5. Then, in Line 6, a constant fraction of the closest lines to $G$ is removed

from the input set $L$. The algorithm then continues recursively for the next iteration, but only on the remaining set of lines until almost no more lines remain. The output is the resulting set $B$.

---

**Algorithm 2:** BI-CRITERIA-APPROXIMATION$(L, m)$

---

**Input:** A set $L$ of $n$ lines in $\mathbb{R}^d$, and an integer $m \geq 1$.
**Output:** A set $B \subseteq \mathbb{R}^d$ which is, with probability at least $1/2$, an $(\alpha, \beta)$-approximation for the $k$-mean of $L$, where $\alpha \in O(1)$ and $\beta = O\left(m^2 \log n\right)$.
`// See Definition 2.5 and Theorem 2.6.`

1  $X := L, B := \emptyset$
2  **while** $|X| > 100$ **do**
3       Pick a sample $S$ of $|S| \geq m$ lines, where each line $\ell \in S$ is sampled i.i.d and uniformly at random from $X$.
4       $G := $ CENTROID-SET$(S)$
5       $B := B \cup G$
6       $X' := $ the closest $7 |X| / 11$ lines in $X$ to $G$. Ties broken arbitrarily.
7       $X := X \setminus X'$
8  **return** $B$

---

**Overview of Algorithm 3.** The input is a set $L$ of lines in $\mathbb{R}^d$, a point $b \in \mathbb{R}^d$ and an integer $k \geq 1$ for the number of desired centers. This procedure is called from Algorithm 4, where $b$ is an approximation to the 1-mean of $L$. The output function $s$ maps every line $\ell \in L$ to $[0, \infty)$, and is being used during Algorithm 4 execution. We define in Line 1 a unit sphere $\mathbb{S}^{d-1}$ that is centered around $b$. Next, in Line 3 for each line $\ell \in L$ we define $\ell' \subseteq \mathbb{R}^d$ to be the translation of $\ell$ to $b$. In Lines 4–5, we replace every line $\ell'$ with one of its two intersections with $\mathbb{S}^{d-1}$, and define the union of these points to be the set $Q$. In Line 6 we call the sub-procedure WEIGHTED-CENTERS-SENSITIVITY that is described in [10]. This procedure returns the sensitivities of the query space of $k$-weighted centers queries on $Q$. As stated in Lemma 32 in supplementary material, the total sensitivities of this coreset is $k^{O(k)} \log n$. Finally, in Line 7, we convert the output sensitivity $s(p)$ of each point $p$ in $Q$ to the output sensitivity $s(\ell)$ of the corresponding line $\ell$ in $L$.

---

**Algorithm 3:** LINES-SENSITIVITY$(L, b, k)$

---

**Input:** A set $L$ of $n$ lines in $\mathbb{R}^d$, a point $b \in \mathbb{R}^d$ and integer $k \geq 1$.
**Output:** A (sensitivity) function $s : L \to [0, \infty)$.

1  $\mathbb{S}^{d-1} := \left\{ x \in \mathbb{R}^d \mid \|x - b\| = 1 \right\}$ `// the unit sphere that is centered at` $b$.
2  **for** $\ell \in L$ **do**
3       $\ell' := $ the line $\{x - b \mid x \in \ell\}$ that is parallel to $\ell$ and intersects $b$ `// see Fig. 3.`
4       $p(\ell') := $ an arbitrary point in the pair $\ell' \cap \mathbb{S}^{d-1}$
5  $Q := Q \left\{ p(\ell') \mid \ell \in L \right\}$
6  $u := $ WEIGHTED-CENTERS-SENSITIVITY$(Q, 2k)$ `// see algorithm overview.`
7  Set $s : L \to [0, \infty)$ such that for every $\ell \in L$

$$s(\ell) := u\left(p(\ell')\right).$$

8  **return** $s$

---

**Overview of Algorithm 4.** The algorithm gets a set $L$ of lines in $\mathbb{R}^d$, an integer $k \geq 1$ for the number of desired centers and a positive integer $m \geq 1$ for the coreset size, and returns an $\varepsilon$-coreset for $L$; See Definition 2.7. In Line 2 a small set $B$ that approximate the $k$-mean of $L$ is computed via a call to BI-CRITERIA-APPROXIMATION. In Line 3 the lines are clustered according to their nearest point in $B$, and in Line 5 the lines sensitivities in the cluster $L_b$ are computed for each center $b \in B$. In the second "for" loop in Lines 8–10 we set the sensitivity of each line to be the sum of the scaled distance of the line to its nearest center $b$ (translation), and the sensitivity $s_b$ that measure its importance with respect to its direction (rotation). Here, scaled distance means that the distance is divided by the sum of distances over all the lines in $L$. The If statement in Line 7 is used to avoid division by zero. In Line 12 we pick a random sample $S$ from $L$, where the probability of choosing a line $\ell$ is proportional to its sensitivity $s(\ell)$. In Line 13 we assign a weight to each line, that is inverse proportional to the probability of sampling it. The resulting weighted set $(S, u)$ is returned in Line 14.

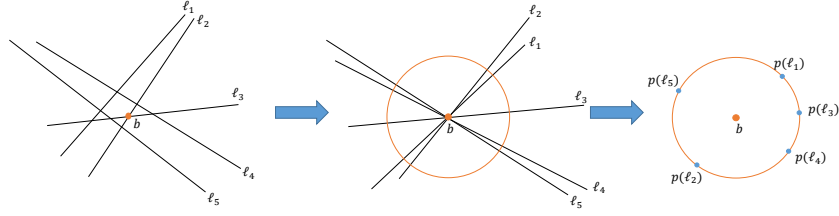

Figure 3: Example of running Alg. 3 in the $d = 2$ dimensional case. On the left, we start with a set $L = \{\ell_1, \ldots, \ell_5\}$ of lines and a single center $b$. Next, we translate each line onto $b$ and stretch the unit sphere $S$ around it. Finally, we replace each line with one of its two intersections with $S$ and achieve the set $Q = \{p(\ell_1), \ldots, p(\ell_5)\}$.

---

**Algorithm 4:** CORESET$(L, k, m)$

---

**Input:** A finite set $L$ of lines in $\mathbb{R}^d$, number $k \geq 1$ of centers and the coreset size $m \geq 1$.
**Output:** A weighted set ("coreset") $(S, u)$ that satisfies Theorem 2.8.

**1** $j := cdk \log_2 k$, where $c$ is a sufficient large constant $c > 0$ that can be determined from the proof of Theorem 2.6.
**2** $B := $ BI-CRITERIA-APPROXIMATION $(L, j)$ // see Algorithm 2
**3** Compute a partition $\{L_b \mid b \in B\}$ of $L$ such that $L_b$ is the set (cluster) of lines that are closest to the point $b \in B$. Ties broken arbitrarily.
**4** **for** *every* $b \in B$ **do**
**5** $\quad$ $s_b := $ LINES-SENSITIVITY$(L_b, b, k)$
$\quad$ // the sensitivity of each line $\ell \in L_b$ that was translated onto $b$;
$\quad\quad$ see Algorithm 3
**6** **for** *every* $b \in B$ *and* $\ell \in L_b$ **do**
**7** $\quad$ **if** $\mathrm{cost}(L, B) > 0$ **then**
**8** $\quad\quad$ $s(\ell) := \dfrac{D(\ell, b)}{\mathrm{cost}(L, B)} + 2 \cdot s_b(\ell)$
**9** $\quad$ **else**
**10** $\quad\quad$ $s(\ell) := s_b(\ell)$
**11** $\quad$ $\mathrm{prob}(\ell) := \dfrac{s(\ell)}{\sum_{\ell' \in L} s(\ell')}$
**12** Pick a sample $S$ of at least $m$ lines from $L$, where each line $\ell \in L$ is sampled i.i.d. with probability $\mathrm{prob}(\ell)$.
**13** Set $u : S \to [0, \infty)$ such that for every $\ell \in S$

$$u(\ell) := \frac{1}{|S| \, \mathrm{prob}(\ell)}.$$

**14** **return** $(S, u)$

---

## 4   Experimental Results

Following motivation to narrow the gap between the theoretical and practical fields, experiments took a dominant place during research.

**Software.** We implemented our coreset construction from Algorithm 4 and its sub-procedures in Python V. 3.6. We make use of the MKL package [26] to improve its performance, but it is not necessary in order to run it. The source code can be found in [1].

**Data Sets.** We evaluate our system on two types of data sets: synthetic data generated with carefully controlled parameters, and a real data of roads map from the "Open Street Map" Dataset [13] and "SimpleHome XCS7 1002 WHT Security Camera" from the the "UCI Machine Learning Repository" Dataset [4]. The roads dataset [13] contains $n = 10,000$ roads in China from the "Open Street Map" dataset (Fig. 4 plot (a)), each road is represented as a 2-dimensional segment that was stretched into

an infinite line on a plane. Synthetic data of $n = 10,000$ lines was generated as well (Fig. 4 plot (b)). **Experiments on offline data analysis.** In Plot (a) in Graph 4, when the sample size was $m = 700$ lines out of 10,000 given lines, the coreset error and variance were 1.86 and 0.16, respectively, that is an error of $\varepsilon = 0.86$, for a sample size of $m = \lceil 602/\varepsilon \rceil$ lines. On the other hand, the error and variance of the competitor algorithm with the same sample size were 2.62 and 0.26. This implies that our coreset is more accurate and stable than RANSAC, and that our mathematically provable constant approximation algorithm for $k$-line mean works better than a standard EM algorithm also in practice. **Experiments on streaming data analysis.** Plot (c) in Graph 4 demonstrates the size of the merge-and-reduce streaming coreset tree during the streaming, which is logarithmic in the number of lines we streamed so far. In Plot (d) in Graph 4, we can see how the coreset construction running time decreases linearly as the number of machines in the machines cluster increases (parameters are written in the chart's title), where coreset construction was measured 3 different times on 3 different number of centers. Note that the decrease rate is almost linear in the cluster's machines number and not exactly, due to overhead of communications and I/O.

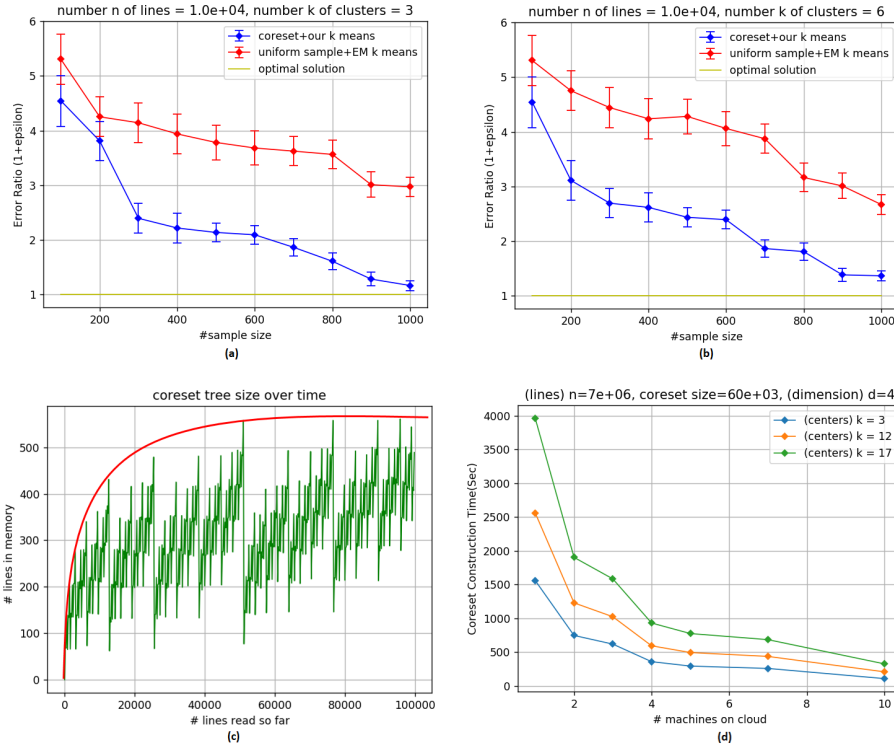

Figure 4: Experiment Results. Graphs (a) and (b) reflects the error decreasing rate in compare to an increasing size of sample by coreset and uniform sampling. Graph (c) shows how the amount of data required by the merge-and-reduce coreset tree is logarithmic in the number of lines read so far in a stream. Graph (d) demonstrate how the coreset construction time decreases linearly in the number of machines in Amazon EC2 cluster, where coreset samples were taken by different number of centers, preserving the invariant.

## 5    Conclusions and Future Work

This paper purposes a deterministic algorithm that computes an $\varepsilon$-coreset of size near-logarithmic in the input. Moreover, we suggest a streaming algorithm that computes a $(1 + \varepsilon)$-approximation for the $k$-means of a set of lines that is distributed among $M$ machines, where each machine needs to send only near-logarithmic number of input lines to the main server for its computations. Other future work will consider an input of $j$-dimensional affine sub-spaces in $\mathbb{R}^d$ (here input of lines is a private case of $j = 1$), where the motivation is multiple missing entries completion.

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
