[Reviews · NeurIPS 2019]

Reviewer 1



K-means clustering for lines is a natural generalization of vanilla k-means problem, and it has potential in dealing with noise, error and missing information. However, few studies have focused on this problem and most of them are either for special cases or heuristic algorithms. This paper proposes the first PTAS algorithm for this problem with dependency near optimal in data size (linear). And the algorithm can also be easily extended to online cases where data may be distributed among several machines. Detailed proof of important results and codes are provided to inspire more follow-up work in this area. Overall structure of this paper is fine and easy to follow, but there still exists some confusion. See improvements for details.

Reviewer 2



The paper is extremely difficult to go read for a non-expert. The sensitivity sampling techniques are made use of in the algorithm design. However, it is hard to grasp the intuition and main ideas behind the algorithms. The algorithms are given as pseudocode and properties of these algorithms are stated in terms of technical theorems. Most of the proofs are in the full version (that I did not go through). I think due to space restrictions, the authors did not have space to give the main ideas behind some of the pseudocode that they give. However, I think it might be better to give the intuition and push the pseudocode to the full version.

Reviewer 3



The authors consider the problem of clustering a set of lines in R^d. The goal is to minimize the k-means objective: given n lines L in R^d find the best set of k points c1,...,ck in R^d so as to minimize sum_{l in L} min_{ci} dist(ci, l)^2. This a clean, nicely motivated problem. The authors provide a coreset construction (namely a small size summary of the input so that any alpha-approximation for the summary yields an alpha(1+epsilon)-approximation for the entire input). This implies the first (1+epsilon)-approximation for the problem with running time nd exp(poly(k)) together with a streaming algorithm with similar running time and memory size 2^{poly(k)} log n. En route to the result the authors provide a bicriteria approximation algorithms: namely a solution that contains O(k (log n dk log k)) centers and whose cost is at most 4 times the cost of the optimal solution with k centers. I think the paper introduces a couple of new techniques and new ideas and make a significant progress on the problem. The ideas behind the approaches (sampling to estimate the location of the center, sensitivity sampling, bounding the VC dimension, merge and reduce, etc.) are not completely new but proving that they indeed work for the case of line clustering is a challenge and a good result. The experiments are ok and seems to indicate that the algorithm is competitive (see comments below though). I think the results are interesting, I recommend acceptance. Comments: - You are saying that you have a deterministic algorithm for the coreset construction, yet the theorem says the opposite. - There are various typos here and there, please check carefully. - Why using EM + kmeans++ and not simply k-means++? - How did you compute the optimal solution? - It seems to me that for getting an offline (1+epsilon)-approximation, one may be able to combined your lemmas on sampling (say Lemma 6.3) together with [A Simple D^2 -Sampling Based PTAS for k-Means and Other Clustering Problems] by Ragesh Jaiswal, Amit Kumar, Sandeep Sen so as to obtain an improved running time of nd 2^k.

[Author Response · NeurIPS 2019]

We thank the reviewers for the not-so-common careful reading.

Following your suggestions:
- The write-up was significantly improved and was sent to a professional English editor.
- Pseudo code was replaced by more intuition and illustrations.
- Python's code was extended with comments and examples on "extreme" synthetic data to give more intuition. As
promised, It will be published upon acceptance, and can be sent to the reviewers already now if necessary.

$k$-Means for lines is much harder than the $k$-means for points which has numerous approximations and coresets. We try
to intuitively explain why below. This is also the reason why it took us few years to write this paper that suggests the
first solution to such a fundamental problem in machine learning. Extension of this explanation was added to the new
version.

### Reviewer 1:

**Q1:** Do you assume that the input space is discrete and bounded?
**A1:** Certainly not. As stated in the main theorems, the guarantees of the algorithms hold for any set of n lines in $\mathbb{R}^d$.
No hidden assumptions.

**Q2:** The objective function of the $k$-means line clustering problem is confusing. In Line 1 it is minimizing sum of
squared distances, which is exactly same as $k$-means. Later on it changes to minimizing sum of distance. I am not sure
this comparison is fair because EM $k$-means is to minimize the sum of squares.
**A2:** This is indeed confusing. The reason is that for simplicity we focused on the classic $k$-means. However, the results
easily generalized to any Lipschitz function of distance, including sum to the power of $z > 0$ or $m$-estimators. Following
this comment, we focus only on squared distances (including experimental results) and moved the generalization to the
last section. We thank the reviewer for this useful comment.

### Reviewer 2:

**Q1:** The paper is extremely difficult to read for a non-expert. Give intuition and move the pseudo-code.
**A1:** Indeed, to obtain such strong provable guarantees we had to use deep mathematical proofs. To help the non-expert:
(1) We accepted the reviewer's suggestion and replaced the pseudo code by illustrations that were added to the overview.
(2) We added detailed comments to our open Python's scripts, as well as example data sets of extreme cases that give
intuition about how and why the algorithms work.

### Reviewer 3:

We appreciate the careful reading of the reviewer.

**Q1:** You are saying that you have a deterministic algorithm but the theorem says the opposite.
**A1:** This is indeed a mistake in the introduction that was fixed. Algorithm 1 is deterministic as claimed in Lemma 6.3,
but our coreset construction holds with high probability as stated in the main theorem.

**Q2:** Why using EM + $k$-means++ and not simply $k$-means++?
**A2:** $k$-means++ was never used in this paper. Unlike EM, we could not generalized $k$-means++ for lines. The reason is
that both its input and output sets are points. In $k$-line means, the input is a set of lines and the output is a set of points.
Moreover, the correctness of $k$-means++ heavily based on metric spaces, while the triangle inequality does not apply
for a set of lines in $\mathbb{R}^d$. For example, constructing a coreset for the case that the optimal cost is 0 is trivial in $k$-means
but not for $k$-line-means.

**Q3:** How did you compute the optimal solution?
**A3:** We use exhaustive search (few days of computation on Amazon's cloud). This is part of our open source code.

**Q4:** It seems to me that for getting an offline $(1 + \epsilon)$-approximation, one may be able to combined your lemmas on
sampling (say Lemma 6.3) together with a Simple $D^2$-Sampling.
**A4:** We aware of this result but unfortunately could not apply it due to the reasons in **A2** above. It would be awesome if
the reviewer can give us a hint in case we missed something.

[Meta-Review · NeurIPS 2019]

This paper proposes an PTAS for k-means clustering of lines. The key contribution is the construction of a small coreset, on which brute force algorithms are run. The authors also extend this to the streaming setting. An important computer vision application is used as motivation. The authors should revise the final version to address the issues raised by the reviewers, and make it more readable to researchers in related but not in the exact area.